# A Parametric Model for the Analysis of the Impedance Spectra of Dielectric Sensors in Curing Epoxy Resins

**DOI:** 10.3390/s23041825

**Published:** 2023-02-06

**Authors:** Alexander Kyriazis, Samir Charif, Korbinian Rager, Andreas Dietzel, Michael Sinapius

**Affiliations:** 1Institut für Mechanik und Adaptronik, Technische Universität Braunschweig, 38106 Braunschweig, Germany; 2Institut für Mikrotechnik, Technische Universität Braunschweig, 38124 Braunschweig, Germany

**Keywords:** cure monitoring, epoxy resin, interdigitated electrodes, film sensor, electrode polarisation, ionic conductivity, ion mobility, dipole relaxation, electrochemical impedance spectroscopy

## Abstract

Observing the curing reaction of epoxy resins is a key to quality assurance in fibre composite production. The evaluation of electrical impedance spectra is an established monitoring method. Such impedance spectra contain the physical effects of dipole relaxation, ionic conduction and electrode polarisation, which shift to lower frequencies as curing progresses. In the early stage of the curing reaction, ionic conductivity and electrode polarisation dominate, and in the later stage of the curing reaction, dipole relaxation dominates. Due to the shift of the effects over several frequency decades, it makes sense to evaluate electrical impedance spectra not exclusively at one frequency but over an entire available frequency spectrum. The measured spectral raw data cannot be easily interpreted by a control algorithm and have to be mapped to simpler key indicators. For this purpose, a frequency-dependent model is proposed to address the aforementioned physical effects. With only five free parameters, measured spectra can be described with a relative error of only 2.3%. The shift of the occurring effects to lower frequencies necessitates switching the key indicator used in the progression of the cure reaction.

## 1. Introduction

Cure monitoring of fibre-reinforced composites offers various advantages. On the one hand, it enables quality management during the production of safety-critical components, and on the other hand, it allows production to be accelerated by reducing uncertainty-based additional times in curing cycles. In the context of a more comprehensive manufacturing monitoring of infusion processes, sensors for in situ cure monitoring have also been used for flow front detection [1,2]. An application in the context of Smart Cure Cycles is also conceivable. Here, the curing epoxy resin is cooled shortly before reaching the gel point to minimise residual stresses by specifically influencing the stress-free temperature of the cured composite [3,4].

The possibilities for cure monitoring are numerous and can be classified in different ways. Cure monitoring can be tool-integrated, component integrated or in the laboratory environment. At the same time several measurands can be used [5], for example, the released heat as in the established calorimetry, mechanical impedance properties [6,7,8], electrical impedances [9,10,11,12] or the occurring reaction shrinkage [13,14]. This article focuses on the component-integrated measurement of dielectric properties implemented by foil sensors with interdigitated structures. On the one hand, the integration of foil sensors enables a very local measurement, but on the other hand, the integration poses various challenges, for example negative influences on mechanical characteristics of the composite due to the introduction of a foreign body [15]. For example, some of these issues are indicated in Figure 1. Our own preliminary investigations have shown that a negative influence of the integrated sensors on the strength properties of the composite can be avoided by a smart choice of the substrate material [16,17]. The substrate material polyetherimide does not only avoid a negative influence but can also exert a limited support effect for adjacent weak metal-epoxy interfaces. When that support effect is taken into account in the sensor design, and the metal shares of the sensor are chosen to be correspondingly low, a sensor can be developed without negative effects on adhesion or crack propagation resistance [18].

In Figure 1, $\underline{Z}=f\left(\omega \right)$ highlights the main issue of this article. The complex electrical impedance $\underline{Z}$ is the measurable property of an integrated sensor with interdigitated electrodes as shown in Figure 1. The impedance depends on the angular frequency ω and the progress of the curing reaction. Cure monitoring with interdigitated sensors is an already well-researched principle, but often, the measured data are only evaluated at a single or, in extreme cases, at only one frequency [19,20,21,22,23,24,25,26]. We propose to describe the measurable impedance spectrum of the sensor by a parametric, physically motivated model. Its model parameters are determined by a curve fit over frequency spectra, and the change in the model parameters over time serves as a metric for cure monitoring. This approach promises a number of advantages: Firstly, fitted parameters are less sensitive to noise than individual measured values because the data reduction during the curve fit involves smoothing along the frequency axis. Secondly, the determined model parameters can be interpreted as a description of actual existing physical effects so that these could be related to the degree of cure with a corresponding material model in further steps. Thirdly, it is prospectively conceivable that influences of the fibre volume content and the resin properties can be separated so that a more comprehensive process observation becomes possible.

The three dominant effects in curing epoxy resin are electrode polarisation, ionic conductivity and dipole relaxation. For high-temperature curing, epoxy resins ionic conductivity is apparent almost over the entire curing range, so correlations between the dielectric loss factor and the degree of cure have already been successfully used [9,10]. In 1994, Maistros and Bucknall already proposed a parametric model that attributes the electrical changes in curing epoxy resin to the three effects mentioned [12]. For modelling ionic conduction and electrode polarisation, they used a model proposed by Day et al. [27] combined with a Cole-Cole-model for dipole relaxation [28], but did not perform parameter extraction from the measured impedance spectra. Own investigations and the work of Cirkel et al. [29] show that the Day model is not well suited to describe electrode polarisation effects. Instead, Cirkel et al. propose a microscopic model based on ionic motions. In 1999, Kim and Char correlated least squares fits of the dipole relaxation time to the degree of cure and the temperature [30]. So far, the most advanced attempt for parameter extraction has been published by Yang et al. in 2021 [31] using least squares fitting on the measured permittivity to identify ion conductivity in situ on an embedded microchip. The degree of cure and the glass transition temperature is then extracted from that information. Electrode polarisation is mentioned in their article but not considered by the fitting model used since it does not strongly influence the observed frequency range.

In this article, a macroscopic equivalent circuit with as few parameters as possible is worked out from the microscopic description to consider ion conductivity and electrode polarisation. This model is combined with a simple dipole relaxation model to form an aggregate model. There are three approaches to modelling the frequency spectra in electrochemical impedance spectroscopy. Firstly, the modelling can be done using complex permittivity. In this approach, the model family around the Havriliak–Negami model is usually used [12,31,32], but these models only represent the effect of dipole relaxation. to model the effect of electrode polarization on the complex permittivity, which is caused by ion migration during the curing reaction of polymers, the Havriliak–Negami model can be modified to account for geometric properties of the electrodes [33,34,35]. Secondly, it is possible to describe the impedance spectrum using an electrical equivalent circuit. This approach allows data reduction and physical interpretability of the data set. This approach has already been used for resin cure monitoring [36,37]. While the relaxation effect can be described by a suitable combination of resistors and capacitors, diffusion effects and electrode polarization effects are described by different types of Warburg and Gerischer elements [38,39,40]. The third possibility is to describe temporal changes in the electrochemical impedance spectrum by means of a superposition of distribution functions. For this approach, there are some models and algorithms for describing diffusion and relaxation effects in electrical impedance spectra [41,42]. However, these do not meet the requirement of physical interpretability, nor does any data reduction take place.

An algorithm and suitable error measure are then proposed to perform parameter fitting of the aggregate model to spectra measured during a curing reaction. This model is applied to a data set recorded on the resin system RIMR426 with the curing agent RIMH435. The explanations in this article are initially limited to curing in pure resin. The average relative deviation between the model and the measured data is of central importance for evaluating the model quality. When interpreting the determined model parameters, it should be noted that they are only reliable within a limited confidence range, which will be addressed in the discussion.

## 2. Materials and Methods

To derive the macroscopic behaviour of the sensor from a microscopic material model, a one-dimensional model is implemented using the finite elements method (FEM). Based on this model, the impedance spectrum of a one-dimensional plate capacitor sensor is predicted by means of limit cycle analysis from the results of transient analysis and an equivalent circuit is set up based on these considerations. The equivalent circuit is then expanded to describe the two-dimensional embedded interdigitated sensor. The model family associated with the Havriliak–Negami model is suitable for modelling dipole relaxation [28,43,44]. The developed overall model is validated with a data set from a real cure measurement. For the comparison between the model and measured data, an averaging relative squared error measure is used, which is suitable for application to complex numbers. By minimising this error measure, the model parameters are determined in a two-step procedure.

### 2.1. Modelling Electrode Polarisation and Ion Conduction

The set of partial differential equations proposed by Cirkel et al. serves as a microscopic material model [29]. The approach works with three fields, namely the electric potential ϕ and the ion concentrations of positive and negative ions $\rho_{\pm }$. The electric potential distribution is modelled with an ordinary differential equation only dependent on location as Poisson’s equation and depends on the boundary conditions and the local spatial charge. The ion motion follows a partial differential equation superimposing a concentration-equalising diffusion motion with an ion motion driven by the electric field. The set of differential equations is given in Equation (Equation 1). The equations for ion motion, thus, correspond to a diffusion equation with an additional force term. Here, the elementary charge *e* is used, assuming that the ions involved are simply positively or negatively charged. *k* and *T* represent the Boltzmann constant and the ambient temperature (21 °C) and together are a measure of the thermal energy. In combination with the impermeability of the electrodes for ions, a concentration or depletion of ions at the electrodes results, depending on the field direction and the type of ions.
(1)$$\begin{array}{cc} \Delta \phi & =-\frac{e}{\varepsilon_{0}\;\cdot \;\varepsilon_{r}}\;\cdot \;\left(\rho_{+}-\rho_{-}\right)\\ \frac{\partial \rho_{+}}{\partial t} & =-\nabla D\left(\nabla \rho_{+}+\frac{e}{k\;\cdot \;T}\;\cdot \;\rho_{+}\;\cdot \;\nabla \phi \right)\\ \frac{\partial \rho_{-}}{\partial t} & =-\nabla D\left(\nabla \rho_{-}-\frac{e}{k\;\cdot \;T}\;\cdot \;\rho_{-}\;\cdot \;\nabla \phi \right) \end{array}$$

The differential equation set is solved using the finite element method for the one-dimensional case. Cubic polynomials are used as shape functions for all three fields to guarantee gradient continuity (C1 continuity) at the element boundaries. This is important for the evaluation of the electric field strength. The region between the electrodes meshes very fine ( 10 pm) in the direct vicinity of the electrodes and coarse in the centre. Using a node distribution algorithm based on the exponential function, the required resolution can be achieved with only 85 nodes. The accuracy of the model is, thus, not limited by numerical effects due to the mesh size but rather by the size of real atoms.

Slow ion movements in the space between the electrodes and fast ion displacements in the immediate vicinity of the electrodes cause the system of differential equations to behave very stiffly, necessitating an implicit formulation. The ion concentrations $\rho_{\pm }$ are implemented spatially uniform at the beginning of the simulation. For epoxy resins, a chloride ion concentration in the range between 4.2 × 10^22^ and 1.6 × 10^24^ Ions m^−3^ is typical [45,46,47,48], for this reason $\rho_{\pm }$ is initialised with 10^23^ Ions m^−3^. The electric potential on the electrodes is fixed by boundary condition to $\phi =\pm \frac{U(t)}{2}$ at the positive and negative electrode, respectively, the electrodes are impermeable to ions. The diffusion constant *D* is set to 10^−14^ m^2^s^−1^. As curing progresses, the diffusion constant is expected to decrease by several orders of magnitude so that the charge carrier mobility also decreases. The time-dependent excitation voltage U(t) is implemented harmonically by means of the cosine function, the time step size is adjusted so that each period of the excitation is resolved with 256 time steps. To determine an impedance spectrum, the model is implicitly solved with harmonic excitation for different frequencies between 10 μHz and 10 kHz. For the determination of the macroscopic impedance spectra, the charge displacement per unit area on the electrodes is first determined using Gauss’ theorem using the permittivity constant ε_0_ and assuming a relative resin permittivity of $\varepsilon_{r}=10$. Since the model is one-dimensional, all charges are related to the unit area:(2)$$\begin{array}{cc} \oint \vec{E}d\vec{A} & =\frac{Q}{\varepsilon_{0}\;\cdot \;\varepsilon_{r}}\\ \;\frac{Q}{A} & =E\;\cdot \;\varepsilon_{0}\;\cdot \;\varepsilon_{r} \end{array}$$

The representation of the time-varying charge displacement over the time-varying excitation voltage is interesting because it results in a Lissajous figure for each frequency. The appearance of these Lissajous figures depends on whether the sensor behaves capacitively or resistively in the frequency range under consideration. Some of the Lissajous figures at the voltage amplitude of U=1V are shown in Figure 2. Due to the cosine function for describing the time response of the exciting voltage, all Lissajous figures start at U=1V and $\frac{Q}{A}=0$. At 10 kHz (bottom right), the Lissajous figure is a straight line, the voltage and the charge displacement are in phase, which corresponds to a capacitive behaviour. At 20 Hz (bottom left), the phase shift between $\underline{u}$ and $\underline{Q}$ is approximately 50°, and the Lissajous figure corresponds to a tilted ellipse. The electrodes, thus, show a mixture of resistive and capacitive behaviour. At 500 mHz, the Lissajous figure is a circle, which corresponds to a phase shift of 90° between charge and voltage and constitutes a resistive behaviour. At 10 mHz, a tilted ellipse is again observed, and for even lower excitation frequencies, the Lissajous figures increasingly deviate from an elliptical shape. The deviation from the elliptical shape is to be interpreted as an expression of non-linear system behaviour. Furthermore, it can be seen in Figure 2 that the system is in an equilibrium state in the frequency range between 500 mHz and 10 kHz during the entire simulation time. In the frequency range between 10 μHz and 10 mHz, a transient system behavior precedes the equilibrium state. In the frequency range between 10 and 500 mHz, a transition to a purely steady-state system occurs. The transient system behaviour arises due to a low frequent interaction of ion concentration at the electrodes with a diffusion-induced smoothing of the ion distribution.

To calculate the impedance spectrum, the amplitude of the fundamental harmonic of the charge displacement is related to the excitation voltage for each frequency point. Since the complex impedance is defined as the ratio of the complex voltage amplitude $\underline{u}$ to the complex electric amperage amplitude $\underline{i}$, the respective excitation angular frequency $\omega_{0}$ is included in the calculation. As the underlying model is one-dimensional, all occurring charge displacements $\underline{Q}$ and currents $\underline{i}$ are related to a unit area and the calculated impedance also includes the unit area.
(3)$$\begin{array}{cc} \underline{Q} & =F\left\{Q\left(t\right)\right\}{|}_{\omega =\omega_{0}}\\ \;\underline{i} & =\underline{Q}\;\cdot \;j\omega_{0}\\ \;\underline{Z} & =\frac{\underline{u}}{\underline{i}} \end{array}$$

Figure 3 shows impedance spectra determined with Equation (Equation 3). These are calculated for different excitation voltages; apart from that, the model parameters are identical to the model parameters for Figure 2. All impedance spectra show a local maximum of the imaginary part of the impedance at about 20 Hz and a local minimum at 300 mHz. Below the local minimum of the impedance imaginary part, the impedance spectra behave differently depending on the excitation voltage. The transition between linear and non-linear behaviour can be found at around 100 mHz and is highlighted by a dashed line in Figure 3. Such an excitation dependence would not appear in a linear system. The non-linearity in the low-frequency range can be attributed to the interaction of diffusion-induced smoothing of the ion distribution with an electrical field-induced concentration of ions at the electrodes. Below the minimum of the imaginary part, non-linear system behaviour is, thus, to be expected.

In the linear range of the system behaviour, this can be approximated by a linear equivalent circuit. This equivalent circuit comprises two parallel branches. The first branch is the capacitance *C*, and the second branch consists of a series connection of the resistor $R_{2}$ and the Warburg impedance $A_{2}$. In the low-frequency range, there are deviations between the simplified equivalent circuit and the model because of the non-linear behaviour. The measured impedance spectra in epoxy resin presented later contain mostly the region above the minimum of the impedance imaginary part. A simplified equivalent circuit is, therefore, sufficient for parameter extraction from measured impedance spectra. The reduction of the diffusion constant with progressing cure leads to a shift of the local maximum as well as the local minimum of the imaginary part to higher impedance values and lower frequencies. This corresponds to an increase in the electrical resistance $R_{2}$ and the Warburg impedance $A_{2}$ in the equivalent circuit. $R_{2}$ in the equivalent circuit is identical with the analytical resistance obtained from the formula $R_{2}\;\cdot \;A=\frac{L\;\cdot \;k_{B}\;\cdot \;T}{e^{2}\;\cdot \;D\;\cdot \;2\rho_{0}}=7.91\;k\Omega m^{2}$.

In contrast to the one-dimensional FEM model considered so far, interdigitated electrodes have a predominantly two-dimensional field distribution. In the area of the fingertips, the field distribution is even three-dimensional. Especially in interdigitated structures with a very large electrode width compared to the gap width, the electric field strengths between the electrodes are considerably larger than the electric field strengths perpendicular to the electrodes. Since the electric field strength is the driving force for ion movement, a distinction must be made between ion movement in the electrode plane and ion movement perpendicular to the electrode plane. For this reason, a third branch is introduced in the equivalent circuit diagram, only comprising a Warburg impedance $A_{3}$. The complete equivalent circuit is shown in Figure 4.

### 2.2. Dipole Relaxation Model

For the description of dipole relaxation, a family of related models exists that describe the frequency dependence of the relative permittivity. These can be derived from the so-called Havriliak–Negami model [44]. However, analysis of impedance spectra yields an electrical capacitance in the first place, not the permittivity. Fortunately, the capacitance *C* and the permittivity $\varepsilon_{r}$ of the epoxy resin are almost perfectly linearly related [19,23], so that the family of models can be applied not only to describe the frequency dependence of the permittivity but also the frequency dependence of the capacitance. Equation (Equation 4) establishes the relationship between the relative permittivity number $\varepsilon_{r}$ and the measurable capacitance *C*. The parameter $K_{IDC}$ contains information about the geometric design of the interdigitated electrodes, and the capacitance $C_{IDC}$ includes the substrate thickness, the substrate material, the geometric design of the interdigitated electrodes, but also the capacitances of the connection line.
(4)$$C=C_{IDC}+K_{IDC}\;\cdot \;\varepsilon_{0}\;\cdot \;\varepsilon_{r}$$

The Debye model, the Cole–Cole model and the Cole–Davidson model can be derived from the Havriliak–Negami model by simplification and are, thus, related to each other to a certain extent, see Figure 5. Each of the above models has a characteristic frequency $f_{C}$. The Debye model has none, the Cole–Cole and Cole–Davidson models have one, and the Havriliak–Negami model has two shape parameters α and β, which take values between zero and one and describe how broad the distribution of the individual dipole relaxation times is. Values close to one mean that the dipole relaxation times have a low spread, and values close to zero stand for a maximum wide spread of the dipole relaxation times. For fixed values, α=β=1, the Havriliak–Negami model transforms into the Debye model.

The Debye model cannot describe the behaviour of the epoxy resin well due to the lack of a free shape parameter, as the epoxy resin has a significant spread of dipole relaxation times. The Cole–Cole model can describe the measured data well. At the same time, the characteristic frequency $f_{C}$ and the shape parameter β can be interpreted independently. In the Cole–Cole model, the characteristic frequency is always the frequency at which the material exactly reaches the mean permittivity between $\varepsilon_{s}$ and $\varepsilon_{\infty }$. Correspondingly the measured capacitance at $f_{C}$ exactly matches the mean value between $C_{s}$ and $C_{\infty }$. Cole–Davidson and Havriliak–Negami models exhibit a coupling between the characteristic frequency $f_{C}$ and the shape parameter α, but are able to describe the measured data well. The coupling causes the characteristic frequency $f_{C}$ to be less easily interpretable than in the Cole–Cole model. For this reason, the constant real capacitance in the equivalent circuit diagram from Figure 4b is replaced by a frequency-dependent complex capacitance according to the Cole–Cole model. The following, thus, applies to the frequency-dependent aggregate impedance:(5)$$\begin{array}{cc} {\underline{C}}_{CC} & =\frac{C_{s}-C_{\infty }}{1+{\left(j\;\cdot \;\frac{f}{f_{C}}\right)}^{\beta }}+C_{\infty }\\ \;\underline{Z} & =\frac{1}{j\omega {\underline{C}}_{CC}+\frac{1}{R_{2}+\frac{A_{2}}{\sqrt{j\omega }}}+\frac{\sqrt{j\omega }}{A_{3}}} \end{array}$$

### 2.3. Fitting Algorithm

A robust fitting algorithm is set up to fit the model to measured data. The algorithm runs in two steps. First, the model parameters for the dipole relaxation are adjusted. Here, the low-frequency capacitance $C_{s}$ and the high-frequency capacitance $C_{\infty }$ are not enabled as optimisation parameters, but only the dipole relaxation frequency $f_{C}$ and the distribution parameter β of the Cole–Cole model. For the fitting algorithm, the impedance data is converted to an equivalent capacitance; see Equation (Equation 6). Only the measured data that is not superimposed by ionic conduction is used for fitting the dipole relaxation behaviour. At the beginning of the curing reaction, this applies only to the higher frequency measurements. With further progress of the curing reaction, it applies more and more to the lower frequency measurements. The measured values for the matching are determined by thresholding at $1.05\;\cdot \;C_{s}$. For the curve fitting, a trust-region-reflective algorithm [49,50] is used, which allows limiting the parameter β to the range between 0 and 1. The absolute squared deviation between the real part of the capacitance of the model and the measured value serves as the error measure for optimisation.
(6)CMeasured=Re1j·2π·Z_MeasuredC*=CMeasured|CMeasured<1.05·C0Error=ReCs−Cinf1+j·ffCβ+Cinf−C*2

In the second step, the ion conductivity and electrode polarisation behaviour with the free parameters $R_{2}$, $A_{2}$ and $A_{3}$ are adapted to the entire spectrum with a Levenberg–Marquardt algorithm [51,52,53]. The capacitance *C* is already described by the dipole relaxation model at this point. To achieve an optimum quickly and reliably, an initial guess is made about the parameter values to offer a starting point for the optimisation algorithm:(7)R2,0=1Re1Z_Measured|fminA2,0=−4π·ImagZ_Measured|fminA3,0=max10·|ZMeasuredf|·2πf

The square of the amount of relative deviation between the measured data and the model is used as the error measure for the optimisation algorithm. The error measure for evaluating the quality of the fit is also derived from this calculated value, which is given in Equation (Equation 8) and is taken up again in the presentation of the optimisation results. Parameter fitting for a single spectrum takes less than 0.1 sec on a conventional computer (Intel i7-9850H processor) and is, thus, sufficiently fast for observing the curing process in real-time.
(8)$$\mathrm{RMSRE}=\sqrt{\frac{1}{N}\sum_{n=1}^{N}|\frac{{\underline{Z}}_{n,Measured}-{\underline{Z}}_{n,Sim}}{{\underline{Z}}_{n,Measured}}{|}^{2}}$$

### 2.4. Cure Experiment

The above model is applied in this article to a measurement data set recorded during the curing of a pure resin sample of RIMR426 resin with RIMH435 hardener in a room temperature environment. For the experiment, an interdigitated electrodes sensor is placed into a 3D-printed mould together with a thermocouple and approx. 7 g of epoxy resin are poured onto the sensor so that it is fully surrounded by epoxy resin. Due to the small resin sample, the increase in resin temperature due to exothermic heat generation is approximately 6 K over the first three hours of cure. The resin temperature stays within a temperature range between 29 and 35 °C over the whole curing experiment. To evaluate the speed of the curing reaction, two isothermal differential scanning calorimetry (DSC) measurements at 30 and 40 °C are carried out.

The film sensor used in the dielectric cure experiment consists of an interdigitated electrode on an approx. 8 μm thin PEI substrate with 30 fingers (15 per electrode), a finger length of 3.7 mm, a finger width of 100 μm and a relatively small gap width of 15 μm. Details on the sensor are given in [54]. The impedance of the sensor is measured in the frequency range between 0.2 Hz and 1 MHz with a Gamry Reference 600+ (Gamry Instruments, PA, USA). 10 measurement points are recorded per decade so that the entire spectrum consists of 68 measurement points.

During the curing process, each 120 s an impedance spectrum is recorded. Taking into account the slow temperature changes and antedating the DSC data shown in Figure 8 the changes of the resin state over one measurement period are negligible. The spectra are shown in Figure 6 as a surface over the time-frequency plane. Since this article is primarily concerned with the fitting model and algorithm, the measurements are considered as input to the algorithm and are, therefore, described here shortly and not in the Results section of this article. At the beginning of the curing reaction, the spectrum is dominated by a pronounced resistive behaviour in the low-frequency range, which shifts to higher impedances and lower frequencies as the curing progresses, until it completely disappears from the observed frequency range. The impedance changes by several magnitudes at this stage of the cure. In the imaginary part of the first recorded spectrum, a pronounced maximum and a pronounced minimum occur in accordance with the frequency response in Figure 3. First, the pronounced minimum disappears from the observed frequency range belonging to the electrode polarisation effect. Later the maximum in the imaginary part disappears from the observed frequency range, mainly belonging to the ion conduction. The aforementioned effect of dipole relaxation is difficult to see in the double-logarithmic plot because, unlike ionic conductivity, it only results in about a doubling of the impedance. It is also worth mentioning the pronounced interferences at 50 Hz, which can be attributed to influences of the power frequency, and an unidentified interference band in the range around 20 Hz. These interference bands are removed from the measurement data not to falsify the determined parameters unnecessarily.

Before the sensor is used for cure monitoring, it is also tested in isopropanol. The purpose of the isopropanol analysis is to investigate the predicted non-linear electrode polarisation behaviour in the low-frequency range. Isopropanol is better suited for this investigation than epoxy resin since its viscosity is lower and the IIMin frequency is, therefore, higher. Measurements in the low-frequency electrode polarisation range, thus, take significantly less time.

## 3. Results

### 3.1. Non-Linearity of the Low-Frequency Electrode Polarisation Effect

The predicted non-linear behaviour of the electrode polarisation effect can also be observed experimentally. Figure 7 shows impedance spectra in isopropanol for different measurement voltages. The measured curves in Figure 7 are qualitatively equal to those in Figure 3. The curves only differ below the imaginary impedance minimum (IIMin), which is around 1 kHz. In the low-frequency range, the course of the impedance measured at a higher voltage is flatter, and the phase angle is closer to 0 ° as in the simulation. In contrast to the simulation, the phase angle returns to −90 ° for even lower frequencies. This indicates that the simulation omits some effects that appear in real electrolytes, but the prediction of nonlinear effects in the low-frequency range is correct. For epoxy resins, however, the IIMin is expected to be much lower than for isopropanol, and nonlinear effects are not expected to have a strong impact on measured spectra.

### 3.2. Calorimetric Curing Process Analysis

Figure 8 shows the investigated resin system’s course of the degree of cure during isothermal cure at 30 and 40 °C calculated from DSC data. During the electrical cure experiment, the temperature lies slightly above 30 °C for most of the curing time, so the course of the degree of cure is expected to run between both curves. Most of the curing happens within the first ten hours in the temperature range under consideration.

### 3.3. Development of Electrode Polarisation and Ion Conduction Behaviour

Figure 9 shows the development of the parameters $R_{2}$, $A_{2}$ and $A_{3}$, which describe the ionic conductivity behaviour as well as the electrode polarisation behaviour. Both the resistance $R_{2}$ and the Warburg impedances $A_{2}$ and $A_{3}$ increase continuously as the cure reaction progresses. The increase in the electrical resistance $R_{2}$ corresponds to a decrease in the diffusion constant *D* on the microscopic level, caused by increased resin viscosity [2,19]. The resin viscosity mainly depends on the degree of cure and on the resin temperature as well.

The two Warburg impedances $A_{2}$ and $A_{3}$ represent the electrode polarisation behaviour, which arises as a result of the interaction of the conductivity of the epoxy resin with the electrodes, which are impermeable to ions. Firstly, the course of the Warburg impedance $A_{3}$ is examined in more detail. The Warburg impedance $A_{3}$ is magnitudes larger than the Warburg impedance $A_{2}$ during the entire curing reaction. In addition, it can be calculated that at any time of the curing reaction and at any frequency considered, the virtual electric current through the Warburg impedance $A_{3}$ is lower than the respective currents in the other two branches connected in parallel. After approx. 3 h, the virtual current through the Warburg impedance $A_{3}$ has become so weak that it can no longer be reliably determined, which is why the values for $A_{3}$ increasingly begin to scatter. Somewhat later, after approx. 4 h, the resistance value $R_{2}$ and the Warburg impedance $A_{2}$ connected in series to it can also no longer be reliably determined, so that the values begin to scatter at a high resistance level. These high resistance values can be interpreted so that the virtual electric current through the $R_{2}A_{2}$ branch is negligible compared to the current through the capacitance. From then on, the curves contain mainly noise and almost no desired signal. The dashed line in Figure 9 indicates the resistance value, at which even at the lowest frequency, the electrical behaviour of the sensor is dominantly capacitive. The spectrum contains so little information about electrode polarisation that the fitting algorithm cannot find a valid solution for $A_{2}$, which is why there is a gap after around five hours in the curve for $A_{2}$.

At the beginning of the curing reaction, the gradient of the electrical resistance $R_{2}$ is lower than the rest of the course. This is because, at the beginning of the curing reaction, the resin sample slightly heats up due to the released reaction heat, resulting in a reduction of the resistance. This temperature effect compensates for the increase in resistance due to the progressive cure at the beginning until the resin sample reaches an equilibrium temperature 5 °C above ambient temperature after approx. 2 h.

### 3.4. Development of the Dipole Relaxation Behaviour

The dipole relaxation is characterised by the parameters $f_{C}$ and β, whose curves are shown in Figure 10. At the beginning of the curing reaction, the cutoff frequency of the dipole relaxation is in the high-frequency range outside the measuring range since the dipoles in the resin are still very mobile. There is not enough information in the measurement data to adjust the parameters $f_{C}$ and β, which is why the values determined cannot be interpreted. After a little more than three hours, the cutoff frequency falls below 1 MHz and is, thus, within the measurement range. There are some disturbances in $f_{C}$ between three and four hours, which can be seen in the curve for the distribution parameter β as well, but in the following four hours, the cutoff frequency decreases approximately exponentially (which appears linear in the logarithmic plot) and falls below 0.2 Hz about seven hours after the start of the experiment. During this time window, the distribution parameter β also decreases from about 0.45 to about 0.15, marking an increasing spread in the dipole relaxation times. These results are consistent with the negative correlation between β and the degree of cure found by Maistros et al. [12].

### 3.5. Considering the Fitting Error

Figure 11 shows the progression of the error measure introduced in Equation (Equation 8). Overall, the mean deviation between the model and measured values ranges mainly between 1.2 to 2.5%. Averaged over all time and frequency points, the RMSRE is 2.3%. The disturbances in the course of $f_{C}$ and β are also recognisable as a slightly increased RMSRE between three and four hours. Due to its mathematical composition, the error measure tends to penalise strong deviations disproportionately more than small deviations so that individual disturbed measurements can strongly affect the error measure as a whole. In practice, the value of the RMSRE can be used to assess the trustworthiness of the parameters found.

## 4. Discussion

### 4.1. Confidence Regions of the Extracted Parameters

The time courses of the parameters extracted from the measurement data shown in the Figure 9 and Figure 10 show that they undergo a mostly smooth and physically explainable change over time. Since all three observed effects shift to lower frequencies during the curing reaction, it is comprehensive that not all parameters are trustworthy in the same way over the entire curing reaction. At the beginning of the curing reaction, the cutoff frequency $f_{C}$ of the dipole relaxation is outside the observed frequency window between 0.2 Hz and 1mHz. *f_C_* is, thus, not a trustworthy parameter for observing the curing reaction until three hours after the start of the experiment. In this time window, the epoxy resin is still in the liquid state and slowly changes to the gel-like state. Therefore, the ionic conductivity and electrode polarisation in the lower frequency range dominate the electrical behaviour and are suitable for observing the cure reaction. With increasing curing, the resistive behaviour retreats to lower frequencies so that the lower frequency range is increasingly dominated by the capacitive behaviour of the epoxy resin.

When a certain resistance value is exceeded, the effect of ionic conductivity has retreated so far to low frequencies that there is no longer enough information in the spectra to reliably identify $R_{2}$ and $A_{2}$. The confidence limit can be considered to be the resistance value $R_{2}=52.5G\Omega$, which is reached about four hours after the start of the curing. This is the resistance value for which capacitive behaviour dominates even at the lowest investigated frequency of 0.2 Hz. After this time, the observation of the curing reaction via the resistance value $R_{2}$ becomes increasingly unreliable for further increasing values of $R_{2}$. The parameter $A_{3}$ loses its trustworthiness even at earlier times but it is irrelevant for cure monitoring anyway.

As soon as the observation of $R_{2}$ is no longer reliable, the observation variable for the cure monitoring system has to be switched to $f_{C}$. Fortunately, the dipole relaxation frequency $f_{C}$ is already in the confidence interval after about three hours. The observer, thus, has to switch to an observation based on $f_{C}$ at some point in the time range between three and four hours. The dominance of the dipole relaxation effect in the spectra is associated with the transition of the epoxy from a gelly to a glassy state [55]. The determined relaxation frequency correlates with the glass transition temperature of the epoxy resin, but also exhibits a temperature dependence [12]. This is, at the same time, an argument for the broadband investigation of the impedance since only a sufficiently wide frequency measurement range can achieve a positive overlap of the observation time windows of $R_{2}$ and $f_{C}$. Restricting the frequency measurement range to the range between 1 Hz and 100 KHz would completely consume the overlap; further restrictions of the frequency measurement range would result in a time window, where no trustworthy parameters are available for observation.

The lower limit of the confidence interval for the parameter $f_{C}$ can be regarded as the lower limit of the measured frequency range at 0.2 Hz, which is reached 7.24 h after the start of the experiment. After this time, no trustworthy parameters are available for observing the curing reaction. By adjusting the frequency measurement range, the observation time window could be extended. However, measurements beyond 100 mHz seem unrealistic since already a single period at the frequency 100 mHz takes 10 s and the measurement of each individual spectrum, thus, takes considerably longer than the 120 s in this article.

### 4.2. Influence of the Nonlinearity of Electrode Polarisation

In any case, the introduction of an equivalent circuit entails a misrepresentation of the real ion conduction behaviour since the latter is clearly non-linear, as can also be read from the descriptive partial differential equation, see Equation (Equation 1). For higher frequencies, this non-linearity is not noticeable since virtually no ion displacement takes place any more, but in the extremely low-frequency range, the non-linearity is clearly recognisable. In Figure 3 and Figure 7, the non-linearity can be seen in the fact that the course of the impedance spectrum is dependent on the excitation voltage.

The boundary between the linear model behaviour and the non-linear model behaviour seems to be the imaginary impedance minimum. For frequencies above the IIMin and approximately one decade below IIMin, all curves in Figure 3 lie on top of each other, and it is possible to describe them with a linear equivalent circuit. For frequencies, more than one decade below the IIMin, a linear equivalent circuit is no longer applicable to the system behaviour since the system behaviour becomes non-linear. This non-linearity also leads to the fact that, strictly speaking, there is no impedance spectrum in the sense of a transfer function since such a function basically requires the linearity of the system. The non-linearity also produces higher harmonics in the current response in the case of harmonic voltage excitation. Therefore, when evaluating measured impedance spectra, the measuring method used to determine this curve must also be taken into account. Impedance curves calculated from a step response are not fully comparable in the low-frequency range with impedance curves determined from a step sinusoidal excitation.

However, the IIMin for epoxy resin is at much lower frequencies than the IIMin for isopropanol and falls below 1 Hz shortly after the start of the experiment. The nonlinearity of electrode polarisation, thus, seems to be a minor problem in the dielectric investigation of epoxy resins.

### 4.3. Enabling and Disabling Parameters for Optimisation

In addition to the five free parameters $R_{2}$, $A_{2}$, $A_{3}$, $f_{C}$ and β, the impedance model also has the parameters $C_{s}$ and $C_{\infty }$, which are fixed in the scope of the investigations presented here. The decision to keep the parameters $C_{s}$ and $C_{\infty }$ fixed has implications for the robustness and computation time of the fitting algorithm, the achievable accuracy and the physicochemical significance of the fitted parameters. Fixing the two parameters makes sense if a significant change in the permittivity of the cured resin is not expected. Although epoxy resins exhibit both some temperature-induced and a curing-induced reduction of the relaxed permittivity $\varepsilon_{s}$ [12,32], this change does not lead to intolerably large deviations between the descriptive model and the measured data. An argument in favour of fixing the parameters is that in many curing stages, $C_{s}$ or $C_{\infty }$ cannot be determined from the measured data because the upper or lower plateau of the permittivity number (i.e., Re(C_)≈Cs or Re(C_)≈C∞) is outside the measured spectral range. In such cases, the measured spectrum does not contain the necessary information to determine reliable values for the parameters. For the consideration of fibre composites, the fixation of both parameters should be at least partially discarded since changes in the fibre volume content also affect the average permittivity in the vicinity of the interdigitated electrodes. Changes of $C_{s}$ and $C_{\infty }$ could, thus, represent valuable information for an extension of the curing monitoring to a more extensive process monitoring. Then, considering the dipole relaxation frequency $f_{C}$ and the distribution parameter β, it has to be decided if $C_{s}$ and $C_{\infty }$ can be calculated from the measured data or if they have to be estimated.

Enabling the parameters $C_{s}$ and $C_{\infty }$ for optimisation requires adjustments to the fitting algorithm. Several attempts with adjusted algorithms reacted considerably more susceptible to noise in the measured data. The more parameters are optimised simultaneously, the higher the dimensionality of the optimisation problem. As a result, the fitting algorithm takes more time and tends to behave more instable. The algorithm used in this article is designed to be simple but robust, and it is not the main issue of the article. If and how satisfactory parameter optimisation of $C_{s}$ and $C_{\infty }$ can be achieved remains an open question for further investigations.

### 4.4. Deducing Microscopic Behaviour from Macroscopic Parameters

Conformal mapping allows relating the complex two-dimensional field course in the environment of the interdigitated electrodes to a one-dimensional field course [56]. The relationship between the permittivity and the capacitance can, thus, be attributed to the relationship in the plate capacitor. Moreover, for the correlation between the specific conductance of the epoxy resin and the measurable electrical resistance, an approximately linear relationship to the measurable resistance value $R_{2}$ can be found. The parameters $C_{IDC}$ and $K_{IDC}$ required for the conversion in Equation (Equation 4) can be determined experimentally by calibration with media of known permittivity number, the parameter $K_{IDC}$ can also be calculated simulatively from conformal mappings. Since the capacitance $C_{IDC}$ also includes the wire capacitances, it can only be determined experimentally by calibration.

A central condition for conformal mapping is that the electrical potential distribution is described by a Laplace equation. In other words, the difference $\left(\rho_{+}-\rho_{-}\right)$ in Equation (Equation 1) must be zero for all locations, i.e., electrode polarisation effects with associated spatial charge zones make the application of conformal mappings impossible turning the differential equation for the electric potential into a Poisson equation. The separation of electrode polarisation ($A_{2}$, $A_{3}$), ionic conductivity ($R_{2}$) and dipole relaxation ($f_{C}$, β) allows to apply conformal mapping to the ionic conductivity parameter $R_{2}$ to determine the specific conductivity of the epoxy resin when $K_{IDC}$ is known. The two parameters $f_{C}$ and β describing the dipole relaxation behaviour are valid for both the micro- and macroscales. The Warburg impedances $A_{2}$ and $A_{3}$ are of little interest for cure monitoring because they result from the non-harmonic transient electrode polarisation process involving a complex interaction of material properties and geometry, which makes it unaccessible for mathematical simplification tools like conformal mapping. For this reason, the relationship between the Warburg impedance and the microscopic properties is not as easy to establish as in the case of the resistance $R_{2}$. However, $A_{2}$ and $A_{3}$ cannot be eliminated from the model, as they are essential for the approximation quality of the model and prevent electrode polarisation effects from distorting the fitted value for $R_{2}$.

### 4.5. Noise and Temperature Dependence

Compared to electrode polarisation and ionic conductivity, which result in impedance changes over several orders of magnitude, the effect of dipole relaxation is much more subtle. At the same time, when observing the curing reaction in an industrial environment, considerably stronger disturbances are to be expected. Due to the high impedance of dielectric sensors and the associated low electrical currents, these sensors are particularly sensitive to electrical interference signals. Thus, it is to be expected that noise will initially impair the observation of dipole relaxation and, thus, the cure monitoring at later times. However, from the gel point onwards, resin shrinkage, which can be measured on film sensors with strain gauges, is also available for evaluating the curing reaction. In a previous article, we, therefore, proposed a foil sensor [54] that combines a dielectric and thermal measurement with a strain measurement. In case of strong disturbances of the dielectric measurement, it is, thus, necessary to switch to the more robust but less informative measurement of the resin shrinkage.

The mentioned temperature measurement of the sensor is not an alternative to the electrical measurement, but provides essential additional information. Without the temperature information, the measurement of non-isothermal cure processes would be impossible because the resistivity of the epoxy resin under investigation is not only strongly dependent on the degree of cure but also on the temperature. An increase in impedance can be explained by both a decrease in temperature and a progression of the curing reaction. As shown in Figure 9 at the beginning of the experiment on the measurement data, both effects can also overlap and produce parameter curves that are difficult to interpret without the temperature information. In industrial curing processes such as the autoclave process, temperature information is often available, but against the background of the slow heat conduction in GFRP laminates, the temperature measurement should be taken locally in the vicinity of the impedance measurement. This reduces the risk of temperature gradients causing the measured temperature and the true temperature around the interdigitated electrodes to differ too much.

### 4.6. Minimum Knowledge Cure Monitoring

The data obtained from the analysis is sufficient to set up a minimal knowledge cure monitoring system, which can detect the vitrification of the epoxy resin but no more properties. Vitrification is often referred to as the dipole relaxation frequency falling below 0.1 Hz [30,57,58] and causes the cure reaction to stop or at least progress much slower. This can be seen after 7.24 h in Figure 10. In this rudimentary variant of cure monitoring, not even temperature information is necessary. All other parameters like $R_{2}$, $A_{2}$ und $A_{3}$ and the conductivity derived from $R_{2}$ via $K_{IDC}$ are unnecessary. They become interesting for more advanced dielectric cure monitoring techniques, which require a resin characterisation to refer to the conductivity and dipole relaxation frequency to the degree of cure. The parameter extraction described here then joins a toolchain with sensor characterisation and resin characterisation, but the setup of that toolchain is not in the scope of the article at hand.

## 5. Conclusions

The presented equivalent circuit proposed in this article can be adapted to the measured data with extremely small deviations by means of an optimisation algorithm. The extracted parameters can then serve as observation variables for a cure monitoring system. In the early phase of the curing reaction, the electrical resistance $R_{2}$, which can also be converted into the specific conductivity of the resin, is suitable for observing the curing reaction. This parameter describes the electrical behaviour of the epoxy resin in the liquid phase up to about the gel point. In the later phase of the curing reaction, the corner frequency of the dipole relaxation $f_{C}$, which correlates with the transition from the gel-like to the glass-like state, is particularly suitable for observing the cure reaction. The correlations between the dielectric material properties and the degree of cure have already been investigated in the literature. Through a comprehensive characterisation of the epoxy resin system used, the model-based dielectric parameters can be related to the resin’s mechanical properties and the degree of cure.

Since the conductivity, in particular, but also the dipole relaxation frequency, are temperature-dependent, the dielectric measurement should be supplemented by temperature measurement. When observing the curing reaction, it should also be noted that the parameter values found are only reliable within a limited confidence interval in each case. The decisive factor for the width of the confidence interval is the width of the considered frequency range. When selecting the measuring frequencies, the largest possible spread between the highest and lowest frequency used should, therefore, be given preference over a fine resolution of the frequency axis. For the epoxy resin system considered here, a change of the observation parameter is unavoidable.

## Figures and Tables

**Figure 1 sensors-23-01825-f001:**
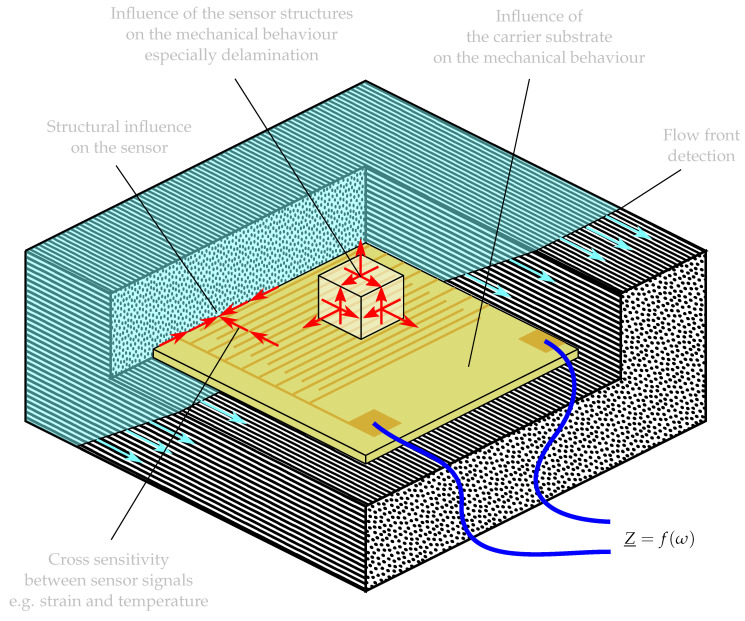
Questions arising from the embedding situation of a film sensor for fibre composite manufacturing observation.

**Figure 2 sensors-23-01825-f002:**
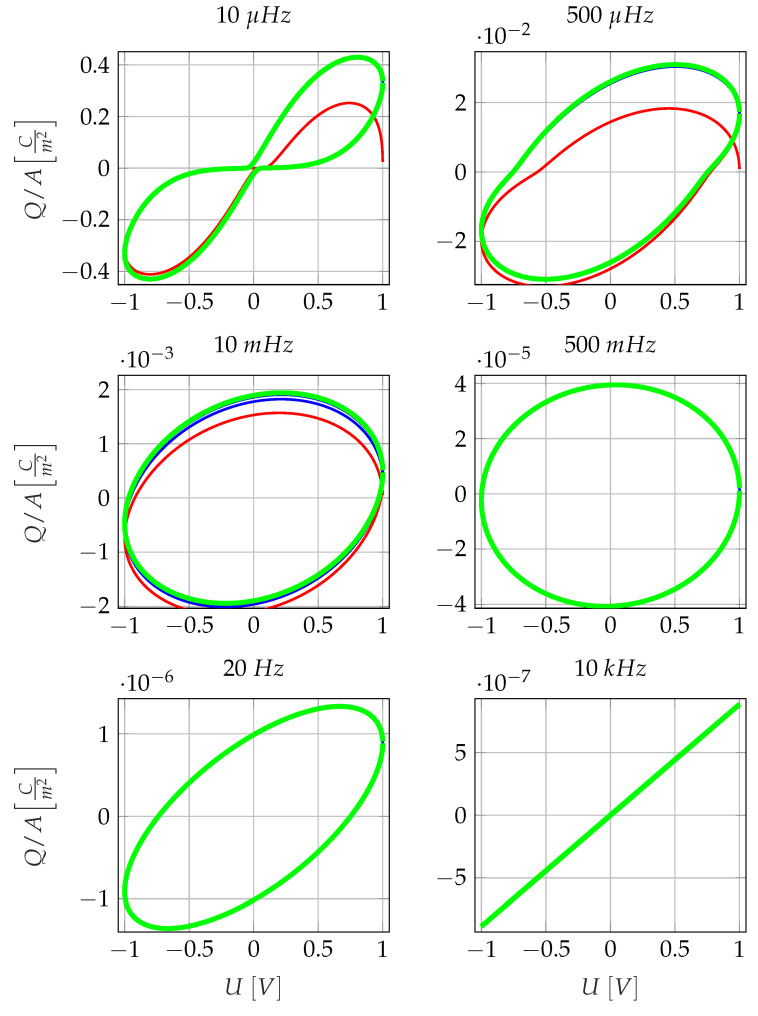
Cyclic charge displacement per unit area $\frac{Q}{A}\left(t\right)$ over cyclic excitation voltage U(t). The parameters for the simulation are as following: excitation voltage U=1V, electrode spacing L=100 μm, diffusion constant *D* = 10^−14^ m^2^s^−1^, average ion concentration $\rho_{\pm ,0}=10^{23}\mathrm{Ions}m^{-3}$. The red lines indicate the initial cycle, the blue lines (where visible) indicate intermediate cycles, and the green lines indicate the fifth cycle, which is identical to the limit cycle.

**Figure 3 sensors-23-01825-f003:**
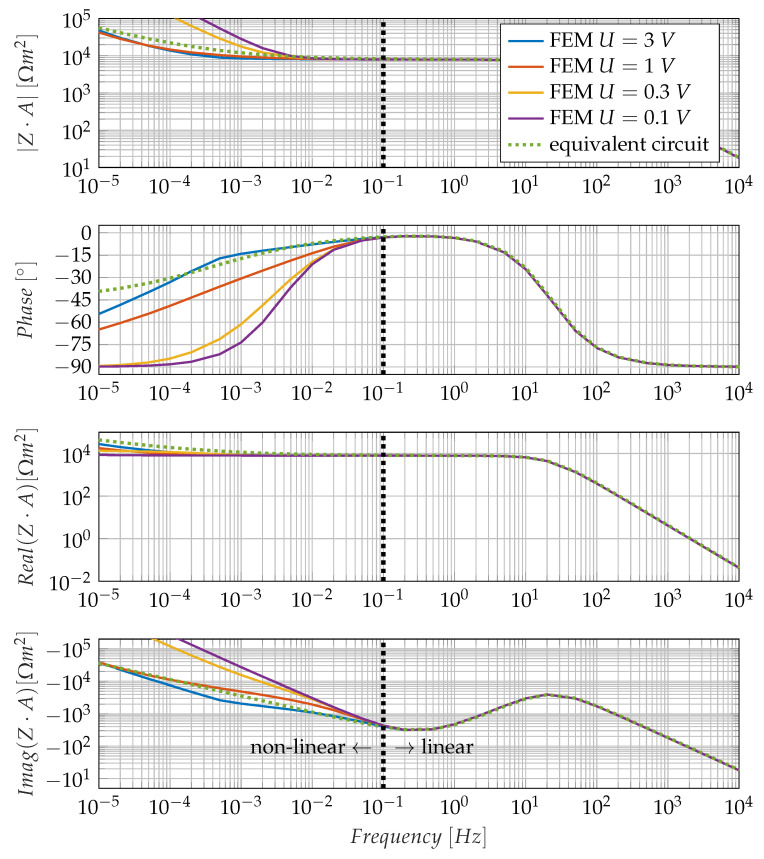
Impedance spectrum of the one-dimensional FEM model. The limit cycles from Figure 2 are marked in the impedance spectrum. The spectrum neglects higher harmonic components and uses the fundamental oscillation of the electrical current for impedance calculation at each frequency. The fitted equivalent circuit uses the parameters $C=860\;nF\;m^{-2}$, $R_{2}=7910\;\Omega m^{2}$, $A_{2}=400\;\Omega \sqrt{Hz}m^{2}$.

**Figure 4 sensors-23-01825-f004:**
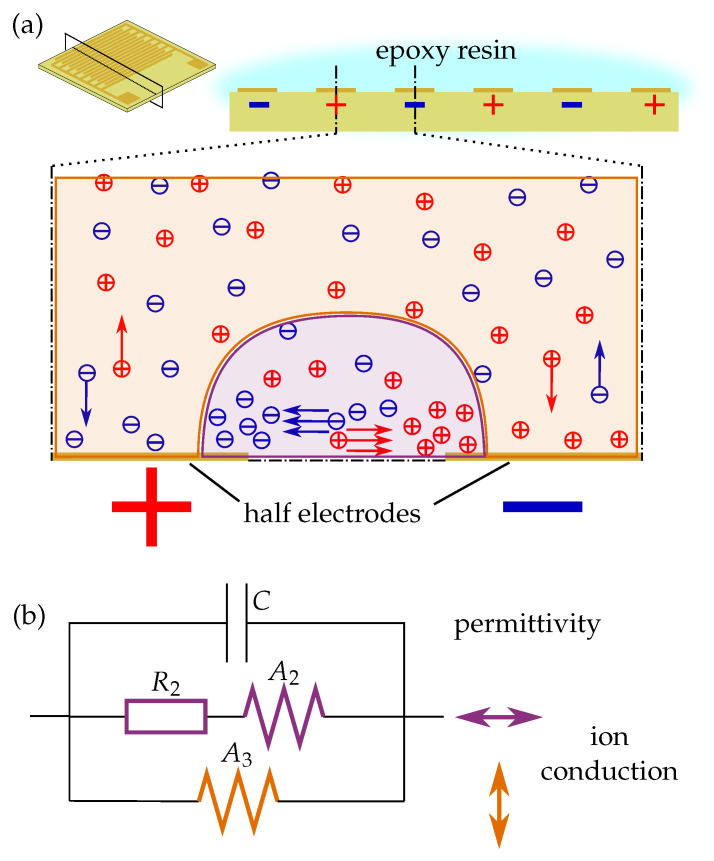
Two-dimensional ion conduction around interdigitated electrodes: (**a**) Illustration of the difference between in-plane conduction and out-of-plane conduction. (**b**) Equivalent circuit for two-dimensional ion conduction and electrode polarisation.

**Figure 5 sensors-23-01825-f005:**
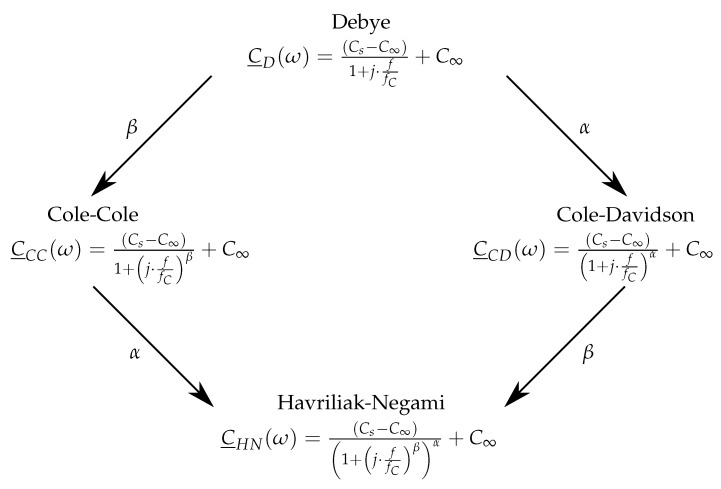
Relationship between dipole relaxation models [28,43,44].

**Figure 6 sensors-23-01825-f006:**
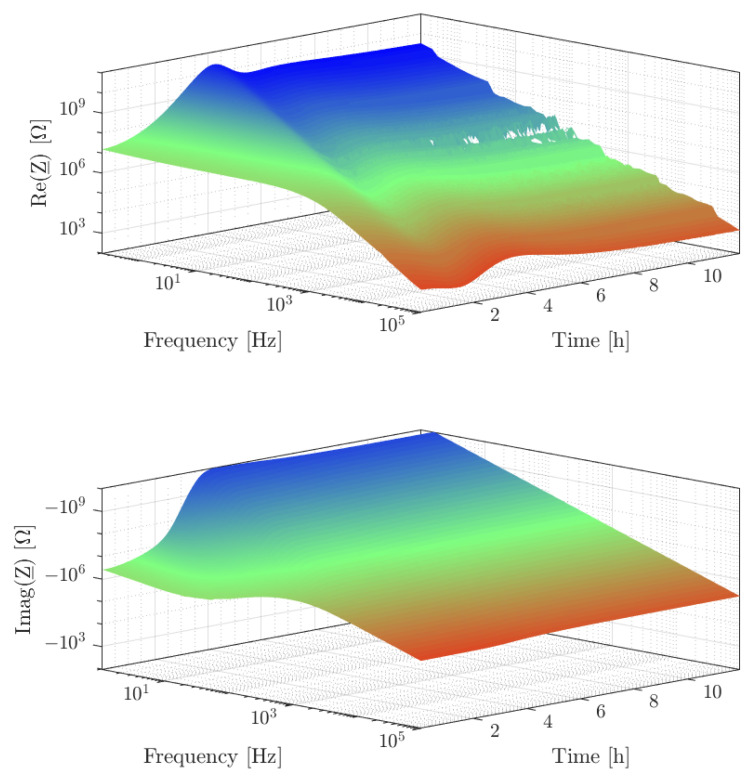
Measured impedance spectra during 12 h curing of pure resin specimen consisting of RIMR426 with RIMH435 hardener. The upper panel displays the real part of the complex impedance, the lower panel displays the imaginary part. The imaginary part is negative because of the capacitive behaviour.

**Figure 7 sensors-23-01825-f007:**
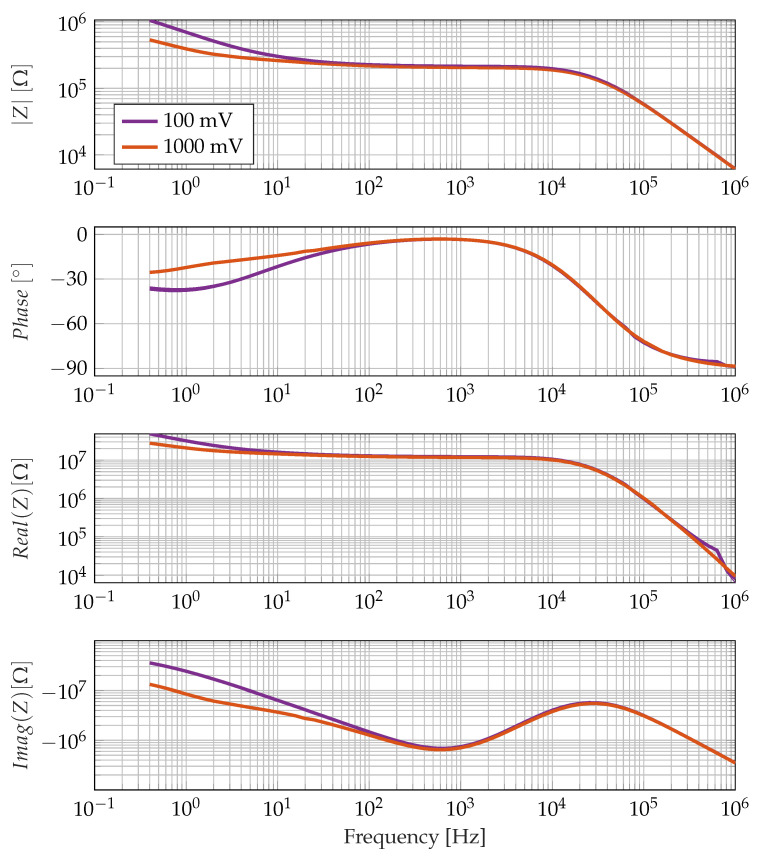
Impedance spectrum of a sensor consisting of interdigitated electrodes in isopropanol at different measurement voltages. For each voltage, the averages of five measured spectra are shown with the standard deviation indicated by thin lines of the same colour. The differences in the low-frequency range are significant.

**Figure 8 sensors-23-01825-f008:**
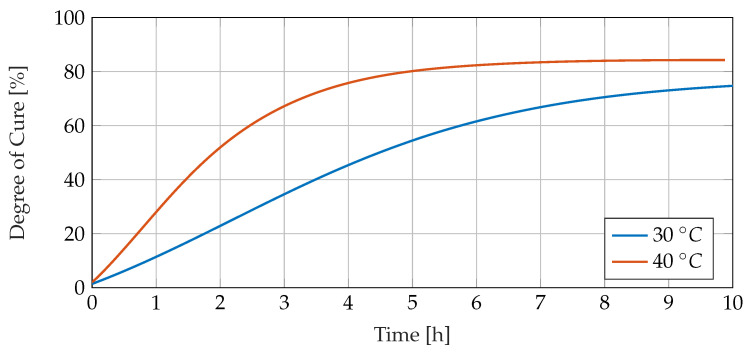
Degree of cure for RIMR426 with curing agent RIMH435 during isothermal cure at 30 and 40 °C calculated from DSC data.

**Figure 9 sensors-23-01825-f009:**
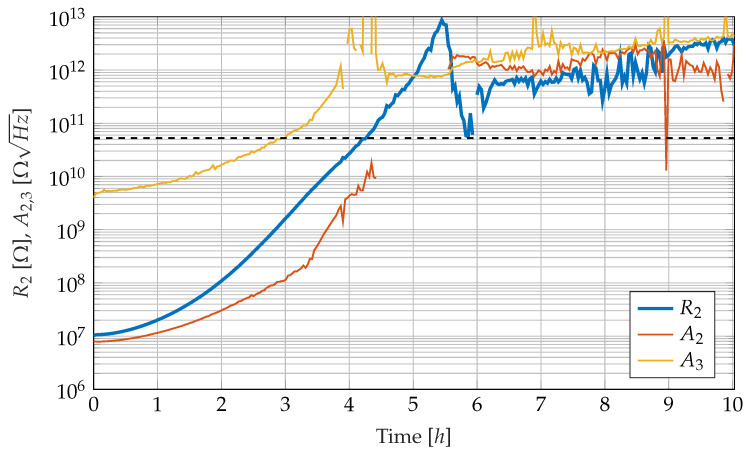
Course of the fitted parameters $R_{2}$ (blue, middle curve), $A_{2}$ (red, lower curve) and $A_{3}$ (yellow, upper curve) describing ion conduction and electrode polarisation. The dashed line indicates the upper border of the confidence range for the parameter $R_{2}$.

**Figure 10 sensors-23-01825-f010:**
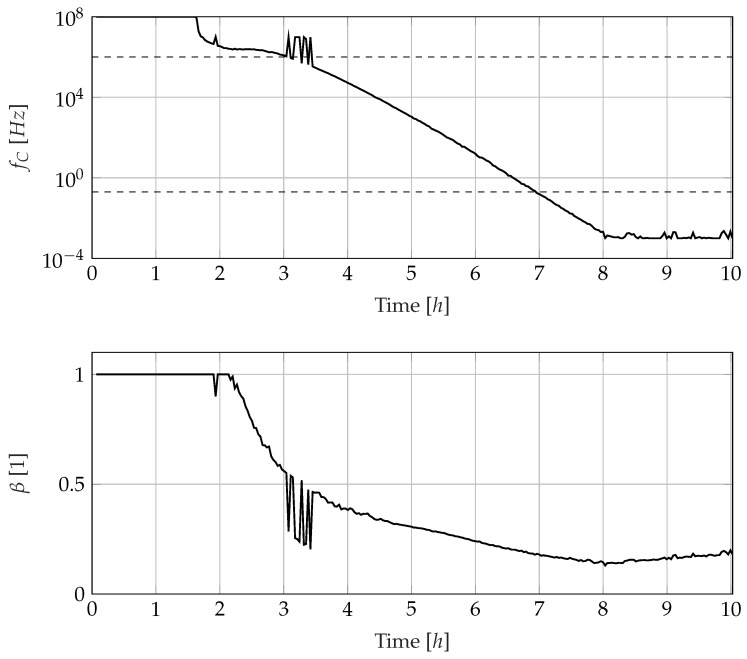
Course of the fitted parameters $f_{C}$ and β describing dipole relaxation.

**Figure 11 sensors-23-01825-f011:**
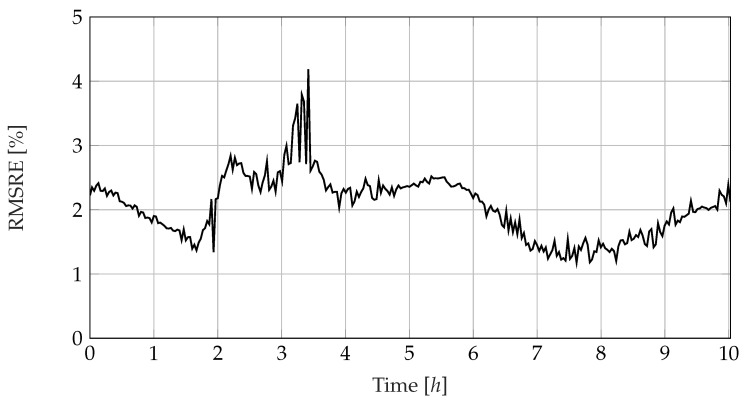
Course of the root mean squared relative error describing the difference between the fitted equivalent circuit model and the measured impedances. The outliers can be attributed to interferences in single measurement sets.

## Data Availability

Measured data and the Matlab code used for parameter extraction can be requested from A.K.

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
