# Peer review of "A Parametric Model for the Analysis of the Impedance Spectra of Dielectric Sensors in Curing Epoxy Resins"

_sensors, 2023, doi:10.3390/s23041825_

Round 1

Reviewer 1 Report

This manuscript models the full frequency response of a thin film sensor in curing epoxy to enable an in-situ cure detection scheme and successfully apply it to FE and experimental data, with some caveats that are explained to my satisfaction in the discussion section. The execution and exposition of this research are excellent and I recommend publication.

A few minor comments follow:

FEM is not defined before usage on line 109

Fig 3 X-axis label Frequenz -> Frequency

On all impedance spectra, consider marking the regions where the raw Lissajous curves indicate linear response, nonlinear response, or draw a line (or thin region) where the response transitions from linear to non-linear.

Consider applying MCMC to the issue of Cs and Cinf. As you say, the paper is not about the fitting algorithm, but a Bayesian uncertainty treatment of a select subset of fits may shed light on why TRF struggles.

Author Response

Dear reviewer,
thank you very much for the positive feedback on our article. In the attached table you can find point-by-point answers to the comments you mentioned.
Kind regards,
Alexander Kyriazis

Reviewer 2 Report

In this paper, a sensor for measuring the curing process of epoxy resin and the corresponding measurement and analysis method are introduced. Some problems need to be improved.

1)The curing process of epoxy resin is an exothermic reaction, so the temperature may be uneven during the curing process. During the measurement, the specific test arrangement and the size of the sensor shall be given to evaluate the measurement results.

2)Temperature is also an important factor affecting the polarization behavior. If the temperature changes sharply or the curing degree changes greatly within a measurement period, the measurement results may be unavailable or special treatment may be required. How did the author consider this problem?

3) Figure 6 is displayed in three dimensions. However, this data presentation method is difficult to distinguish some locations.

4) Epoxy resin is thermosetting material. Will the sensor mentioned in the article be fixed in the epoxy resin after curing, causing damage to the structure of the epoxy resin?

5) What is the purpose of using Isopropanol analysis? The physical state of epoxy resin changes with the curing reaction, while Isopropanol is always a viscous liquid. There is no comparison between the two materials.

6) The whole measured change process is closely related to the curing process of epoxy resin. It is suggested that the author measure and analyze the curing process of the measured formula system itself. And compare with the measurement results of this paper to verify the accuracy of the measurement.

Author Response

Dear reviewer,
thank you very much for the feedback on our article. You addressed some points we definitively had to make more clear. You can find the answers to your
comments as well as the list of changes in the attached table.
Kind regards,
Alexander Kyriazis

Reviewer 3 Report

The manuscript is very well written and addresses a significant problem of the epoxy curing mechanism, which can help to prepare better epoxies. However, it is less relevant to the journal "Sensors". The manuscript can be offered a transfer to "Electrochem" or "Polymers". 

This study is very interesting to understand the curing process which involves multiple chemical reactions and the English language is of a very good standard.

References are appropriate and figures are of good quality and all text in them is readable.

Results are very well presented and supported by the data. The manuscript can be accepted without any revisions. 

Author Response

Dear reviewer,
thank you very much for the feedback on the article. Since you did not propose to make changes on the manuscript there is no appendix with a list of changes.
Kind regards,
Alexander Kyriazis